# Surface Modification of Additively Fabricated Titanium-Based Implants by Means of Bioactive Micro-Arc Oxidation Coatings for Bone Replacement

**DOI:** 10.3390/jfb13040285

**Published:** 2022-12-08

**Authors:** Anna I. Kozelskaya, Sven Rutkowski, Johannes Frueh, Aleksey S. Gogolev, Sergei G. Chistyakov, Sergey V. Gnedenkov, Sergey L. Sinebryukhov, Andreas Frueh, Vladimir S. Egorkin, Evgeny L. Choynzonov, Mikhail Buldakov, Denis E. Kulbakin, Evgeny N. Bolbasov, Anton P. Gryaznov, Ksenia N. Verzunova, Margarita D. Apostolova, Sergei I. Tverdokhlebov

**Affiliations:** 1Faculty of Medicine and Health, Harbin Institute of Technology, Harbin 150080, China; 2School of Nuclear Science & Engineering, Tomsk Polytechnic University, 30 Lenin Avenue, 634050 Tomsk, Russia; 3Institute of Chemistry of FEB RAS, 159 Pr. 100-letiya Vladivostoka, 690022 Vladivostok, Russia; 4Cancer Research Institute, Tomsk National Research Medical Center, Russian Academy of Sciences, 5 Per. Kooperativny, 634050 Tomsk, Russia; 5Roumen Tsanev Institute of Molecular Biology, Bulgarian Academy of Sciences, Acad. G. Bonchev Str., bl. 21, 1113 Sofia, Bulgaria

**Keywords:** bioactive coatings, calcium phosphate coatings, micro-arc oxidation, additively manufactured metal implants, X-ray computed tomography, biocompatibility

## Abstract

In this work, the micro-arc oxidation method is used to fabricate surface-modified complex-structured titanium implant coatings to improve biocompatibility. Depending on the utilized electrolyte solution and micro-arc oxidation process parameters, three different types of coatings (one of them—oxide, another two—calcium phosphates) were obtained, differing in their coating thickness, crystallite phase composition and, thus, with a significantly different biocompatibility. An analytical approach based on X-ray computed tomography utilizing software-aided coating recognition is employed in this work to reveal their structural uniformity. Electrochemical studies prove that the coatings exhibit varying levels of corrosion protection. *In vitro* and *in vivo* experiments of the three different micro-arc oxidation coatings prove high biocompatibility towards adult stem cells (investigation of cell adhesion, proliferation and osteogenic differentiation), as well as *in vivo* biocompatibility (including histological analysis). These results demonstrate superior biological properties compared to unmodified titanium surfaces. The ratio of calcium and phosphorus in coatings, as well as their phase composition, have a great influence on the biological response of the coatings.

## 1. Introduction

In recent decades, additive manufacturing (AM), such as 3D printing technologies, are widely used in medical practice [1,2,3,4,5,6]. The use of these technologies allows the creation of implants with the required size and shape, catering to individual requirements of any patient, which was impossible by means of conventional production methods [7,8]. Other indisputable advantages of AM are the reduction of production materials waste, such as in the form of splints required for top down manufacturing processes, as well as the high geometrical and dimensional accuracy of manufactured products (e.g., for polymeric 3D printers ~50 µm, with other processes reaching higher resolutions [9]). In the future of orthopedics and traumatology, 3D printing of metals will play an important role, as it provides the ability to form metal frames with different internal hollow structures (voids) [10,11,12]. Varying the internal void configuration provides the ability to adjust the mechanical strength of the product, reducing implant weight and utilizing less source material [11,13]. Further, it allows for an increase in surface area, enabling new bone tissue to grow deep into the implant, which enhances its fixation in the patient’s body [11,13]. A main issue with 3D printed implants is the remaining powder in the case of selective laser sintering, which can hinder cell growth and, therefore, must be dissolved prior to implantation [14]. The widespread use of implant structures created by additive technologies poses new challenges for quality control of their structure and modification of their surface.

One of the methods to meet the challenges in quality control is X-ray computed tomography (X-ray CT). It is an effective non-destructive quality control method for analyzing the final produced geometry, uniformity, dimensions and internal elemental distribution of complex 3D structures [15]. This method further allows the evaluation of the product at multiple length scales, ranging from macro to micro levels, with the ability to build 3D visualization models [16,17]. Furthermore, the possibility of evaluating the uniformity and thickness of internal coatings over the entire product volume is a significant advantage [18]. With the performance increase in new detectors, novel beam guiding progress, IT-based data treatment and computational power, as well as developing research protocols, the speed and information content of X-ray CT studies continues to increase [19,20].

Highly novel 3D printed structures still often lack crucial functions [8]; therefore, there are currently new methods for functionalizing metal implants, even with anti-bacterial coatings, under investigation [21]. One of the most promising methods for modifying the surface of complex 3D products with biocompatible coating structures is micro-arc oxidation (MAO), also known as plasma electrolytic oxidation (PEO) [12,22,23]. As a rule of thumb, articles devoted to MAO of metal alloys for biomedical applications study, generally, the effect of electrochemical parameters on the morphology and resultant coating properties [24,25]. MAO has a number of important technological advantages, including cost-effectiveness, simplicity, environmental safety and high throughput of the method [26,27]. Coatings formed by this method are characterized by increased surface wear resistance [28], antibacterial properties [29], corrosion resistance [30] and high adhesion strength to the substrate [31]. The porous morphology of MAO coatings promotes better adhesion of the implant to bone tissue, and the relatively large surface area provides a large number of attachment points for cell growth [32]. In addition, the porous structure can serve as a reservoir for loading various drugs into MAO coatings [33,34,35]. Since the properties of MAO coatings mainly depend on the composition of the electrolyte, substrate, process parameters and processing time, the development of this method is commonly carried out in two directions: modification of the composition of the electrolyte solution and increasing the power of electrical equipment [27,36,37,38]. The focus of most MAO-related work is the modification of the electrolyte solution composition [38].

Until today, there is no research and no complete comprehensive analysis of the coatings formed in various electrolytes and, particularly, the comparison of oxide coatings regarding their biocompatibility. The need for this study is particularly important, as some calcium phosphate MAO coatings are already used in orthopedic practice. The novelty and uniqueness of this study is based on a fully comprehensive investigation of the composition of electrolyte and similar MAO coating times on the internal and surface structural parameters of the resultant coating. Here, despite similar coating times, the structure and physical parameters of the coatings, in terms of porosity, differ vastly. This effect yielded a comparable amount of coating material, causing the most isolating coating to be the thinnest layer. Such a result is presented for the first time. Such physical differences of the same coating time are compared for the first time, in terms of biocompatibility in *in vitro* and *in vivo* studies, whereby previous studies were, in terms of production time and coating layer total material, not comparable. In addition, the applicability of X-ray computed tomography (X-ray CT) combined with a software aided analysis to study the structural quality of the obtained MAO coatings deposited on additively manufactured 3D Ti-6Al-4V implants is demonstrated. X-ray CT was applied due to the ability to evaluate the uniformity, thickness and structural quality of MAO coatings, especially on the internal parts of the 3D implants.

## 2. Materials and Methods

### 2.1. Coatings Fabrication

Three different sample types made out of Ti6Al4V alloy conforming to ASTM F136 (Standard Requirements for Wrought Titanium-6-Aluminum-4-Vanadium Ultrafine Grain Alloy for Surgical Implant Applications (UNS R56401)) were coated by the “micro-arc oxidation” (MAO) process. Sample discs with a diameter of 10 mm and a height of 1.5 mm allowed for physico-chemical, mechanical and *in vitro* investigations of the MAO coatings. Larger discs composed out of the same materials with a diameter of 40 mm and a height of 1.5 mm were utilized for electrochemical corrosion tests. Implant samples that were 3D printed with a mushroom-like shape were used for X-ray tomography and *in vivo* tests (dimensions are shown in Appendix A). Samples in the form of discs were cut from titanium rods by electrical discharge machining. These titanium rods were purchased from the company “VSMPO-AVISMA Corporation, PSC” (Moscow, Russia). The mushroom-like shaped 3D implant samples were fabricated using selective laser sintering (SLS) by the company “LCC Logeek^s^MS” (Novosibirsk, Russia).

The formation of three different MAO coatings on the three sample types were performed using the MAO setup “Micro-arc oxidation complex”, which were designed in the Laboratory for Plasma Hybrid Systems at “The Weinberg Research Center”, School of Nuclear Science and Engineering, Tomsk Polytechnic University (Tomsk, Russia). This MAO setup is based on a pulse generator according to the patent [39].

All experimental sample types used were arranged into three groups in terms of their coating composition, which was formed by one of the three different electrolyte solutions used in this study. The MAO coating process is discussed in detail in the Appendix A. In Appendix A, the macroscopic appearance of the three different sample types with the three different MAO coatings applied on the three different sample types is shown.

### 2.2. Coating Characterization

Roughness of the coatings was measured via contact profilometry on a profilometer (Talysurf 5–120, Taylor Hobson, Leicester, UK). The sample roughness is characterized by the root mean square roughness (R_a_) and the “ten-point height of irregularities” (R_z_). Surface structure and morphology were characterized by a high-resolution scanning electron microscope (Supra 55, Carl Zeiss AG, Oberkochen, Germany) equipped with a Raith 150TWO electron beam lithography system under a high vacuum (4.8 × 10^−6^ mbar) in secondary electrons (SE) mode. In this analysis, the accelerating voltage and beam current were set to 1.5 kV and 74 pA, respectively. The working distance was 6 mm. Energy-dispersive X-ray spectroscopy (EDX) analysis was carried out by an EDX spectroscope (JSM-5900LV, JEOL Ltd., Tokyo, Japan) under high vacuum with an accelerating voltage of 30 kV; the beam current was 0.12 pA, and the working distance was 15 mm. Coating thickness measurements were carried out by using a thickness meter KONSTANTA 5 (LLC “KONSTANTA”, Saint-Petersburg, Russia). Wettability of the sample surfaces has been measured on a drop shape analyzer (Easy Drop DSA 20, Krüss, Hamburg, Germany) using the Drop Shape Analysis software, v1.92.1.1 (Krüss, Hamburg, Germany). Therefore, the contact angles of water (Solopharm, Saint Petersburg, Russia) and diiodomethane (99%, Acros Organics, Geel, Belgium) were measured. Five droplets with a volume of 3.0 µL each of each liquid were placed on the sample surfaces to obtain the water contact angles (WCA) and diiodomethane contact angles (DCA). The results were used to calculate the surface energies applying the Owens–Wendt–Rabel–Kaelble (OWRK) method. Coating thickness measurements on top of the 3D printed implants were acquired by a customized X-ray tomograph designed at the Tomsk Polytechnic University (Tomsk, Russian Federation, for details as well as image analysis method, see Section 2.3 in this chapter).

Investigation of the coating crystal structures was performed with an X-ray diffractometer (XRD 6000, Shimadzu, Kyoto, Japan) using monochromatic CuK-alpha (1.540598 Å) radiation utilizing an accelerating voltage of 40 kV and a beam current of 30 mA. The XRD executed measurements covered a scanning angle range from 10° to 80° (*θ*–2*θ* geometry), with a set scan speed of 2°/min, a sampling pitch of 0.02° and a signal integration time of 1 s. The scratch test was performed by a nanoindenter (G200, Agilent’s Electronic Measurement, Santa Clara, CA, USA). The scratch tests were carried out as follows: a triangular Berkovich pyramid shaped nanoindenter was scratched with a linear increase in the load from 0 mN to 100 mN for a distance of 400 µm utilizing a constant speed of 10 μm/s over the surface. Each test consisted of three scratches for each sample followed by microscopic investigation of the coating integrity by SEM. The indentation hardness (H_IT_) and indentation elastic modulus (E_IT_) of the coatings were measured by a DUH-211S dynamic ultra-micro hardness tester (Agilent’s Shimadzu, Tokyo, Japan). Indentation was performed with a triangular pyramid indenter (opening angle 115º) at 50 mN load on 10 positions for each sample type. H_IT_ was defined according to the ISO14577-1. Indentation places for MAO 1 and MAO 3 were chosen by an optical microscope so as to avoid getting the indenter into pores. Statistical analysis was performed with Statistica 7.0 (StatSoft, Tulsa, OK, USA) for evaluation of the data, by one-way analysis of variance and a Mann–Whitney U test. The data presented in this article are displayed as average value with mean standard deviation.

Electrochemical properties of the surface layers were investigated by an electrochemical measurement station composed of four VersaSTATs 3 units equipped with the Frequency Response Analyzer (FRA) option (VersaSTAT MC, Princeton Applied Research, Oak Ridge, TN, USA). Measurements were carried out in a three-electrode cell K0235 (purchased from the same company) in 0.9% aqueous NaCl solution with a platinized niobium mesh-based counter electrode and a K0265 Ag/AgCl-based reference electrode. A 0.9% NaCl solution is commonly used as a corrosive solution for examining the degradation behavior of biocompatible materials, since 0.9% NaCl is comparable to the ionic strength in mammals [40]. The exposed sample surface area was equal to 1 cm^2^. Prior to the electrochemical measurements, the samples remained for 60 min in the solution at open circuit potential (OCP) in order to reach a steady state. The electrochemical measurement station recorded the potentiodynamic polarization (PDP) curves at a scan rate of 1 mV/s [41]. Polarization started from −30 mV versus *E_C_* (corrosion potential) and increased at a scan rate of 0.167 mV/s up to +1700 mV. Recorded polarization curves (working electrode potential E vs. current density j) were fit by Levenberg–Marquardt (LEV) approach utilizing the following equation [42,43]:(1)$$j=j_{C}\cdot \left(10^{E-E_{C}/\beta a}+10^{E-E_{C}/\beta c}\right)$$

The values that provided the best fit of the corrosion potential *E_C_*, the corrosion current density *j_C_*, the slope of the cathodic polarization curve *β*_c_ and the slope of the anodic polarization curve *β_a_* were identified. The electrochemical impedance spectroscopy (EIS) measurements utilized a sinusoidal perturbation signal with 10 mV r.m.s. amplitude, with a set frequency range from 0.1 MHz to 0.01 Hz and logarithmic sweep (10 points per decade) at OCP. The experiments for the determination of impedance characteristics and polarization behavior were carried out in the following order: OCP, EIS, OCP, PDP. In order to avoid distortions, the polarization resistance values were determined from separate experiments using another sample from the same coating type. The linear polarization resistance (LPR) experiments started from minus 30 mV and increased at a scan rate of 0.167 mV/s up to plus 30 mV against the OCP. Linear potential-current density plot as the *Rp* = Δ*E*/Δ*j* allowed for calculation of the polarization resistance *Rp* [44]. Three different analytical software systems utilizing different algorithms were used to investigate the experimentally obtained data (VersaSTUDIO, Oak Ridge, TN, USA; ZView and CorrView, both from Scribner Associates Inc., Southern Pines, NC, USA). These experiments were carried out for three times on the large titanium disk sample type.

### 2.3. X-ray Computed Tomography (X-ray CT)

Measurements of the thickness and uniformity of the fabricated coatings on the 3D printed titanium implant samples were performed with a self-made X-ray microtomography scanner (micro-CT), which has been designed and assembled in the international research laboratory for X-ray optics at the Tomsk Polytechnic University (Tomsk, Russia). Micro-CT micrographs of the coated samples were received by a customized X-ray scanner utilizing a Hamamatsu microfocus X-ray source (L9181-02, Hamamatsu Photonics, Hamamatsu, Japan), with a tube acceleration voltage of 130 kV and a current of 80 µA, without beam filtering [45]. A dynamic high-resolution X-ray flat panel (Mark2430C, PRODIS.TECH, Ovrazhki, Moscow, Russia) was used for image acquisition [46]. The voxel (grid value in a 3D room) size is 4.7 µm at a geometric magnification of 18× and 2.1 µm at 40× magnification. An image acquisition time of 900 ms per scanner position was set, resulting in a total scan duration of approximately 15 min for 1000 projections. The reconstruction of the tomograms was performed using a standard back projections algorithm with flat field, ring artifacts and beam hardening correction. In order to analyze the 3D tomograms in a fully automated manner and to plot of the thickness distribution by size, a software for analysis and visualization of CT data was used (VGSTUDIO MAX 3.4.5, Volume Graphics GmbH, Heidelberg, Germany) utilizing the module “Wall Thickness Analysis” [47].

### 2.4. Cell Line Utilized for In-Vitro Tests

Adult human adipose-derived cells MSCs (hADMSCs) were isolated from liposuction waste in Cancer Research Institute of Tomsk National Research Medical Center of Russian Academy of Sciences (Tomsk, Russia) (permission no. 11 on 23 October 2015, a scan of this permission in Russian and a translation into English are available as Appendix A) following the methodology described in reference [48]. Cells were re-suspended in a complete nutrient medium DMEM/F12 (Gibco, Waltham, MA, USA) with 10% fetal calf serum (Hyclone Laboratories Inc., Logan, UT, USA) and antibiotics: penicillin 5000 units/mL and streptomycin 5000 µg/mL (Paneco, Moscow, Russia). Cells were incubated at 37 °C and 5% carbon dioxide in a humid condition.

### 2.5. Cell Viability Test (MTT Assay)

For this test, 3-(4,5-dimethylthiazol-2-yl)-2,5-diphenyltetrazolium bromide (MTT, Paneco, Moscow, Russia) was prepared as a stock solution of 5 mg/mL in phosphate-buffered saline (PBS, cell culture grade, pH 7.2, Paneco, Moscow, Russia) and filtered. hADMSCs were plated in 96-well plates at a density of 1 × 10^5^ cells per sample. At the days 1, 3 and 5, a solution of 20 µL of MTT was added to each well. After incubation for 4 h at 37 °C, the medium solution was discarded and 100 µL dimethylsulfoxide (DMSO, Paneco, Moscow, Russia) was added to each well, and the absorption was measured by a plate reader Multiscan FC (Thermo Fisher Scientific Inc., Waltham, MA, USA) at 570 nm to determine the cell viability. In this test, viable cells reduce MTT to the purple-colored formazan compound, whereas this reaction will not take place in the dead cells [49].

### 2.6. Cell Adhesion

Cell adhesion was determined using fluorescent staining. After incubation of 1 × 10^5^ cells per sample in a 48-well plate (day 1 and day 5), the cells were fixed with 2% formaldehyde (pure for analysis, Reahimlab, Moscow, Russia) and stained with antibodies against β-catenin (Sigma-Aldrich, Burlington, MA, USA), following the manufacturer’s instruction. As a secondary antibody, Alexa488-anti-mouse was used (Sigma-Aldrich, Burlington, MS, USA). In order to stain the cell nuclei, 4′,6-Diamidin-2-phenylindol (DAPI, Invitrogen, Waltham, MA, USA) was used. The fluorescent staining of the cells was determined using EVOS M7000 imaging system (Thermo Fisher Scientific Inc., Waltham, MA, USA), acquiring 10 images per sample and calculating the cell density.

### 2.7. Alkaline Phosphatase Detection

Alkaline phosphatase (ALP), as a marker of osteogenic differentiation, was determined on day 21 after cell application to the test samples, according to test protocol of the manufacturer QuantiChrom™ using an Alkaline Phosphatase Assay Kit (BioAssaySystems, Hayward, CA, USA). Therefore, 1 × 10^5^ cells were placed on the surface of each sample in a 48-well plate. Optical density was measured at wavelength of 405 nm using a Multiskan FC microplate photometer (Thermo Fisher Scientific Inc., Waltham, MA, USA), and ALP concentration was calculated as international unit per liter (IU/L).

### 2.8. Alizarin Red Staining

Alizarin Red (Sigma-Aldrich, Burlington, MA, USA) was applied for staining the cells to investigate osteogenic differentiation, according to reference [50]. Therefore, the cells were fixed with 10% formaldehyde (cell culture grade, Reahim, Moscow, Russia) and then incubated for 5 min in a 2% Alizarin Red solution. Thereafter, the cells were washed three times with distilled water and evaluated microscopically using an inverted microscope (EVOS M7000 imaging system, Thermo Fisher Scientific Inc., Waltham, MA, USA) and analyzed with Image J V1.52a (National Institutes of Health, Bethesda, MD, USA) [51,52].

### 2.9. In Vivo Test

Twenty California rabbits of the oryctolagus cuniculus breed (6 male rabbits; 14 female rabbits, 2.3–2.5 kg, 3 months old at the beginning of the experiments, veterinary certificate number 6934022345 of 23 September 2020, purchased from KFH Kurilenok, Novo-Kuskovo, Tomsk Oblast, Russia) were used to implant the 3D printed implants (a scan of the ethic commission allowance for this study in Russian and a translation into English are available as additional Appendix A). The *in vivo* experiments were carried out in accordance with the rules adopted by the European Convention for the Protection of Vertebrate Animals used for Experimental and Other Scientific Purposes (ETS 123, 18 March 1986, Strasbourg, France). Depending on the types of test objects, the rabbits were divided into three groups (5 rabbits in each group):(1)first group—rabbits implanted with 3D printed Ti implants surface-modified with MAO 1 coatings (control group);(2)second group—rabbits implanted with 3D printed Ti implants surface-modified with MAO 2 coatings;(3)third group—rabbits implanted with 3D printed Ti implants surface-modified with MAO 3 coatings.

The group with a titanium dioxide coating was used as a control group, since implants with such a coating are widely used in clinical practice. A bone defect spanning 9 × 9 mm^2^ in size with a depth of at least 5 mm was formed in the region of the nasal bone along the midline of each animal using a surgical motor system attached to a high-speed spherical cutter (to place the cylindrical part of the implant in it). In almost all cases, connection with the nasal cavity was noted. The wounds were washed with an aqueous solution of chlorhexidine. Subsequently, the sterilized 3D printed implants with three different coatings were placed into the area of the formed defect. It was ensured that the cylindrical part of the implant fitted tightly into the formed bone hole in the nasal bone; the flat part of the implant was placed extramedullary on the bone surface. Photographs of the formed bone defect and fixation of the 3D printed implant are shown in Appendix A. The implants were fixed on the edge of the bone defect using 2 self-tapping miniscrews (length—5 mm, width—1 mm) manufactured by Conmet LLC (Moscow, Russia). Subsequently, the wound was washed again with a 0.05% chlorhexidine (Merck KGaA, Darmstadt, Germany) solution. After fixation of the implants, a layered wound closure was performed to prevent edema. Animals were placed in cages with food and water ad libitum. After waking up from anesthesia, the animals restored, in all cases, their natural activity within 15–20 min. Animals were euthanized on the 60th day of the experiment by a toxic dose of carbon dioxide. When the implant was excised, the location of the implant, contact with tissues, and the absence of suppuration were assessed macroscopically. Next, tissues were taken for morphological examination. The fragments of the facial bones (nasal bone) were resected with a fixed implant and fascial tissues covering the bone. Tissue samples (together with the implant) were placed in bottles with a solution of 10% formaldehyde (analytical grade, Reahimlab, Moscow, Russia) for fixation, with the ratio of fixative solution volume to sample being 20:1, for at least 1 day. The preparation of slices was carried out in the Department of the General and Molecular Pathology of the Cancer Research Institute of the Tomsk National Research Medical Center of the Russian Academy of Sciences (Tomsk, Russia).

The bone fragments were decalcified in 8.5% formic acid solution (purity for analysis, purchased from CJSC “Vekton”, St. Petersburg, Russia). After that, the tissue samples were dehydrated and impregnated with paraffin in a Leica ASP-300S tissue processor (Leica Biosystems Nussloch GmbH, Nussloch, Germany). Further, the material was mounted on cassettes. The slices with a thickness of about 5–7 µm were prepared on a Leica RM 2255 rotary microtome (Leica Biosystems Nussloch GmbH, Nussloch, Germany), mounted on glass slides and stained with a solution of hematoxylin and eosin (purchased from JSC “LenReaktiv”, St. Petersburg, Russia) in a Thermo Gemini AS histology slide stainer (Thermo Fisher Scientific Inc, Waltham, MA, USA). Subsequently, the microscopic examination was carried out using a light microscope (Axio Scope A1, Carl Zeiss Microscopy Deutschland GmbH, Oberkochen, Germany). The study includes an analysis of the contact between the implant material and bone tissue, assessment of the intensity of inflammatory process and composition of the inflammatory infiltrate, assessment of the intensity of the fibroblast reaction, and the presence and degree of osteogenesis. A summary of the used sample types, surface modification process via MAO, the applied measurement principles and setups for investigations is graphically displayed in Figure 1.

## 3. Results

Surface morphology: The morphology of the fabricated coatings highly depends on the chemical composition of the electrolyte and the MAO process parameters [26,36,38,53,54]. Figure 2 displays the morphology of the coatings fabricated by the micro-arc oxidation method in different electrolytes. Determined MAO 1 surface morphology is crater-like, in the center of which there is a channel/pore (Figure 2A,D) with an average pore diameter of 0.90 ± 0.56 µm and 34.5 ± 1.5 pores per 1000 µm^2^ (Figure 3A and Appendix A). MAO 2 displays rougher structure and is represented by hollow spherical particles and pore containing hemispheres (Figure 2B,E and Figure 3A). In the case of these samples, two Gaussian peak distributions for pore sizes, as well as spheres, are determined, whereby the number of pores per 1000 µm^2^ with an average size of 10.71 ± 2.50 µm is 19.6 ± 2.4, while the number of spheres per 1000 µm^2^ with an average diameter of 3.76 ± 1.08 µm is 25.6 ± 2.3. Upon optical investigation, the surface morphology of the MAO 3 coatings resembles the one of MAO 1 (Figure 2A,C), with the only difference being the number and size of pores (Figure 2D,F and Figure 3A). The number of pores on the surface coating of MAO 3 is almost five times greater than that of the MAO 1 (34.5 ± 1.5 vs. 172.5 ± 12.6). MAO 3 pores have an average diameter of 0.82 ± 0.20 µm, which is close to MOA 1. Such a porous structure is common for MAO coatings, which is due to micro-discharges occurring during the MAO process [27]. Although the pores can negatively affect the anti-corrosion and wear-resistant properties of the coatings, such a porous structure has a positive effect on the attachment of cells to the surface for biomedical applications [55,56,57]. In the present case, the arithmetic average of the roughness profile (R_a_) and the maximum roughness (R_z_) of MAO 1 and MAO 3 coatings have approximately the same values (Figure 3B and Appendix A). MAO 2 coatings have rougher surfaces than the ones of MAO 1 and MAO 3. Another effect of plasma discharge activities is the formation of micro-cracks on the surface (Figure 2D,F). Isolated shallow micro-cracks were determined on the surface of all examined coatings (Figure 2D–F). The wettability and surface energy results are given in Appendix A and Appendix A. As can be seen, the applied MAO coatings are significantly increasing the samples’ wettability for water. Especially MAO 2 coatings are showing the highest wettability by polar liquid (water) but also the lowest wettability by non-polar liquid (diiodomethane). In terms of the surface energy, they are higher compared to the unmodified control samples. A gradual increase in surface energy can be observed from MAO 1 to MAO 3. For the disperse and polar component, this trend is not present. Here, the MAO 2 coating shows the lowest dispersive component of the surface energy, and the MAO 1 coating shows the lowest polar component.

Analysis of cross-section SEM micrographs allows a precise determination of the coating thickness, as well as the internal coating structures. With 3.9 ± 1.2 µm, the smallest coating thickness and the lowest number of pores are observed for the MAO 1 coating (Figure 2G and Appendix A). In contrast, the MAO 2 coating has with 42.0 ± 2.1 µm the highest coating thickness of any MAO coatings fabricated and has the largest pores with sizes varying from 1 µm to 5 µm (Figure 2H and Appendix A). The MAO 3 coatings with 21.9 ± 5.5 µm are almost half as thick as the MAO 2 coatings (Figure 2I and Appendix A). The SEM micrographs clearly show that the thickness of this coating is not uniform, and only a few pores are visible on the cross-section SEM micrographs. The sizes of these pores range from 1 to 3 µm (Figure 2I). Figure 3A displays an overview of the average pore number and their average diameter for the three MAO coatings.

Elemental composition: Figure 3C and Appendix A display the elemental composition of the MAO coatings investigated by EDX. These EDX measurements reveal compositions typical for titanium oxide and calcium phosphate containing coatings. The main difference between MAO 2 and MAO 3 calcium–phosphate coatings are their calcium to phosphorus (Ca/P) ratio. MAO 3 coatings are composed of a higher amount of calcium than phosphorus and reveal, therefore, a higher Ca/P ratio than MAO 2 coatings (Figure 3C and Appendix A).

The Ca/P ratio can be a useful indication of the higher biological activity of these MAO coatings in the process of bone regeneration. It is worth noting that not only the elemental composition but also the distribution of the elements differs significantly between the coatings MAO 1–3 (Figure 4). Despite its relatively thin layer thickness, MAO 3 in particular has a layered structure compared to MAO 2 (Figure 4), which indicates different biological properties for MAO 2 and MAO 3 coatings. This layered structure becomes particularly clear when several elements are superimposed, as graphically displayed in Appendix A. This layered structure is a result of the MAO fabrication process for MAO 2 and 3, as well as the ion composition of the solution, as outlined in detail in reference [27]. The ionization of titanium at the interface forms a dense layer of titanium dioxide, followed by a porous layer of calcium phosphate. Due to the absence of these ions, in the case of MAO 1, only a titanium dioxide layer forms, resulting in the thinnest and most dense coatings of the three investigated sample types.

Phase composition by X-ray diffraction (XRD): The phase composition of the examined coatings by XRD reveals that MAO 1 and MAO 3 coatings contain four phases: titanium dioxide—anatase and rutile, α-Titanium (α-Ti) and amorphous phase (Figure 3D,E and Appendix A, Appendix A). MAO 1 and 3 coatings share similar composition of the crystallite phase fractions, while MAO 2 is nearly exclusively of amorphous composition. For all three MAO coatings, the amorphous fraction is dominating. Presence of phase of the alpha titanium [58] for all three MAO coatings is explained by the titanium substrate composition. In this context, it is important to note that for all samples, a significant amount of signal related to amorphous phases is detected, especially for the MAO 2 and MAO 3 coatings. This signal correlates with loosely ordered structures in the nanometer range and sizing up to a few to dozens of nanometers. Such an interpretation is particularly in line with the small angular scattering of X-rays for surface features, according to Yoneda [59] and the general theory of neutron and X-rays scattering on interfaces [60]. This interpretation is supported by SEM micrographs revealing features that can easily be observed in Figure 2. Especially the shell thickness in Figure 2E and the pore structures in the coatings in Figure 2G,I support this rationale. For a detailed discussion of individual peaks and reflection planes, see Appendix A in the Appendix A.

Scratch test and nanoindentation: The scratch test is a common, fast and effective method to evaluate adhesion properties of coatings [61]. Analysis of the SEM micrographs of all MAO coatings investigated after their scratch test reveals a plow groove (Appendix A, upper line). A scratching of the coatings by an increasing nanoindenter load up to 100 mN did not allow for reaching the underlying substrate by the indenter. Only a ploughing of the coating material with the formation of the plow grooves along the scratching lines (Appendix A, profilograms) were achieved. This result proves the high stability of the film, and the absence of any de-lamination proves high degree of anchoring to substrate (Appendix A). The elastic moduli for MAO 1 and MAO 3 are comparable, while MAO 2 is significantly softer (Appendix A). For a detailed discussion on ploughing effects, see Appendix A. The penetration depth of the intender is in case of MAO 1 and MAO 3 significantly lower compared to MAO 2 (Appendix A). This hints to a significantly lower elastic modulus for MAO 2. Such a rationale is supported by Appendix A, in which MAO 1 has the highest elastic modulus followed by MAO 3, which displays the second highest modulus. As Appendix A demonstrates, MAO 1 and MAO 3 are mostly homogeneous, with the grooving occurring due to the plowing of the coating surfaces by nanoindenter, whereby the intender performs a nearly linear increase in penetration depth. It is worth noting that the two zones are not obvious in the scratch test SEM micrographs in the case of MAO 3, since there are no significant differences in mechanical properties and a very thin first layer coating. This rationale is also supported by XRD, which does not display additional phases (Figure 3D). On the other hand, MAO 2 demonstrates clear differences in the form of a significantly softer coating and thus a higher degree of penetration per time unit, which is also supported by the large amorphous phase composition determined by XRD (Figure 3D). This effect is most likely due to the eggshell-like structure of the MAO 2 coating. In this case, it should be noted that the samples tested are not polished or smoothed to prevent falsification of these measurements. Thus, the scratch test results look rather noisy, which is due to the structural peculiarities of the samples.

Corrosion test: Since the biomedical application of coatings implies their constant exposure to body fluids, their corrosion resistance is an important property, along with their morphology, elemental and phase composition [29]. For this reason, electrochemical studies by electrochemical impedance spectroscopy (EIS) were performed. Electrochemical studies demonstrate that all samples exhibit a high convergence, hinting at efficient MAO coating of the large disc samples. This proves the reliability of the applied technological modes of MAO coating formation. Analysis of the measured values and a comparison with the calculated fits equivalent electrical circuit (Figure 3F–H and Appendix A, Appendix A) show that the coatings significantly protect the underlying metal. MAO 1 yields the highest level of protection among all of the examined coatings (Figure 3G,H). For the MAO 1 coating, the calculated corrosion current density (Appendix A) is 5, 7 and 10 times lower compared to MAO 2, MAO 3 and the unmodified titanium alloy sample, respectively. In the passive region of the potentials at more than +1 V, the ratio of measured currents remains the same for the coated samples MAO 2 and MAO 3, but reaches 1000 times for MAO 1 and the sample without coating (Figure 3F–H).

This result can be explained by modelling a theoretical equivalent circuit composed out of a two-step internal layering, a porous outer layer and a thin non-porous tight barrier layer close to the titanium substrate (Appendix A). The lowest Q2 (Q is the parameter of the constant phase element (CPE) of an EIS measurement [62]) value among all MAO coated samples for MAO 1 (Appendix A) indicates an effective barrier layer. Another simpler explanation of the barrier layer of the MAO 1 coating is that the outside pores are simply not interconnected, which is supported by SEM micrographs in Figure 2. This could explain the Q2 parameter values in Appendix A. Simulations of the EIS spectra of the MAO coatings are not only possible via an equivalent electrical circuit containing two R-CPE chains (Appendix A), but theoretical diffusion limitations are also possible. Such an assumption accordingly leads to a Warburg diffusion element (a CPE that is independent from the frequency) within the circuit, which allows for a high agreement between simulated and experimental data (Figure 3G,H).

X-ray computed tomography (X-ray CT) investigations: Figure 5 shows microtomography images of the scanned complex-shaped 3D implant samples with MAO coatings. Since the formation of MAO coatings on the surface of 3D printed structures was carried out in MAO modes, different from ones applied for flat samples (for more details see Appendix A), the thickness of the coatings on the 3D printed samples and titanium discs differs significantly. Coating studies using X-ray CT demonstrated a non-uniform coating growth on the surface of complex-shaped 3D printed products during the MAO formation process. In Figure 5, the layer thickness distribution on the MAO coated 3D printed implant samples is shown. Here, the highest thickness level is indicated by blue colored dots. According to the theory of the MAO process [27,63], a dielectric breakdown occurs when a certain critical voltage is exceeded. The current flows only in the breakdown areas, which leads to a local thickening of the coating at these areas. If this voltage is not sufficient for a dielectric breakdown, the formation of coatings at these areas does not occur. A new dielectric breakdown occurs at a different location with a thinner coating layer where the resistance to voltage difference will be less, and, thus, the voltage is sufficient for a breakdown. Therefore, dielectric breakdowns occur at different areas on the surface of a sample. When the coating thickness becomes the same over the entire surface of the sample, and the applied voltage is no longer sufficient to form a coating of higher thickness, the subsequent growth of the coating can only occur with an increase in the applied voltage. The diagrams in Figure 5D,H,L display the results of the unimodal MAO coating thickness distribution over three regions of interest with a radius of 1 mm and a height of 0.7 mm. In the case of MAO 3, the coating does reveal a normal distribution. The average thickness of the MAO coatings according to the presented diagrams is (57.59 ± 1.50) µm for MAO 1 coatings, (41.74 ± 2.08) µm for MAO 2 coatings and (59.79 ± 8.52) µm for MAO 3 coatings. Implant sample surfaces-modified with MAO 2 coatings are characterized by the smallest coating thickness distribution (Figure 5). It is worth noting that no defects from the SLS printing process were found in the bulk volume of the titanium substrate.

Cell adhesion and proliferation: Figure 6D displays bright-field micrographs of adherent mesenchymal stem cells (MSCs) after 21 days of cell cultivation, displaying no detrimental effects even after prolonged exposure time. Fluorescence micrographs in Figure 6E,F show adherent MSCs on the surface of the examined samples obtained after 1 day and 5 days following plating. In all fluorescence micrographs it can be observed that the cells have a regular spindle shape with a well-defined membrane. These cells were automatically counted by the image analysis software of the EVOS M7000 imaging system (Figure 6A). Until day 5, the number of adherent cells increased on all sample surfaces, which means that the cells are well proliferating and are able to form a dense monolayer. On the first day, the number of adherent cells was higher on both control (−) (plastic Petri dish) and unmodified titanium alloy samples (Ti control (+)) compared to the samples with the MAO 1 and MAO 2 coatings (Figure 6A). The largest amount of adherent cells was observed on the MAO 3 coated samples (Figure 6A). After 5 days, all samples showed high cell densities and similar values for all investigated samples (Figure 6A,F). These results prove the absence of cytotoxic effects of the MAO coatings on MSC cell adhesion and proliferation. It is noted that in MAO 2 fluorescence micrographs the cells appear smaller and rounder, as they are partially located in coating pores and/or on the top of the eggshell-like structure. The increase in cell density within the first hours shows that MAO 3 promotes the best early adhesion of all three samples. As the wettability results in Appendix A and Appendix A are showing, the lowest wettability for water has been found for the control sample, as this sample also shows the lowest cell proliferation. The MAO 3 coatings are exhibiting the highest values for the cell proliferation and low contact angles for both the water contact angles (WCA) and the diiodomethane angles (WCA).

*In vitro* test for cell viability (MTT assay): An MTT assay determines the cell viability utilizing the reduction of 3-(4,5-dimethylthiazol-2-yl)-2,5-diphenyltetrazolium bromide (MTT) to the purple-colored formazan in living cells. As shown in Figure 6B, starting from day three, significant differences between control (−), Ti control (+), as well as MAO coated samples, are evident. The increased cell viability (day 5) on the surface of calcium phosphate containing MAO coatings (MAO 2 and MAO 3) compared to the titanium oxide MAO 1 coating demonstrates the improved biocompatible properties of these MAO coatings, proving the microscopic observations. The dispersive surface energy component γ^D^ results shown in Appendix A and Appendix A are increasing, which indicates a correlation for the cell proliferation, especially for day 1 of this test (Figure 6A). This finding is in agreement with a recent publication on surface energy and cell proliferation [64].

Cell differentiation: Since the prepared MAO coatings are intended to be utilized as bone implants, the osteogenic differentiation of the MSCs is of central interest. Therefore, the osteogenic differentiation of the MSCs was measured by the alkaline phosphate (ALP) activity causing formation of calcifications after 20 days of cell cultivation (Figure 6C) [65]. The ALP activity levels on day 20 for the MAO 1 coating are comparable to the control (−) samples, while the ALP levels for MAO 2 and MAO 3 are two times higher compared to control (−) and MAO 1 coating samples (Figure 6C). The lowest osteogenic differentiation activities were observed for unmodified Ti control (+) samples without MAO coatings (Figure 6C). Calcification formation proves that the highest amount of cells with osteogenic differentiation can be observed for the MAO 3 coated samples, which is also proven by Alzarine Red stained samples (Figure 6D and Appendix A). From the present results, it is obvious that all investigated samples support the adherence of MSCs and show no cytotoxic properties, which is in agreement with the microscopic investigations and MTT measurements. In addition, the MAO 3 coating offers the best conditions for stimulating the osteogenic differentiation of MSCs followed by MAO 1, and the titanium control, whereby MAO 2 is displaying relatively low calcification (Appendix A and Appendix A).

*In vivo* study evaluation: After implanting the mushroom-shaped 3D printed implants surface-modified with the MAO 1–3 coatings at the nasal cavity of the rabbits, the implants were maintained in the live rabbits for 60 days. Subsequently, the rabbits were euthanized and the implants with connected bone tissue removed (Figure 7). The external examination of the implantation site and the surrounding tissues revealed only in a few isolated cases signs of inflammation. In all cases, the investigated 3D printed implants were covered with connective tissue and firmly fixed to the bone tissue by utilizing titanium-based mini-screws (Figure 7A–C). Implants with MAO 1 coating were covered with a fairly pronounced connective tissue that penetrated into the pores of the implant, meaning that the implants were in close contact with the bone tissue (Figure 7A,D). The connective tissue covering the implants with the MAO 2 coating was more pronounced than on implants with the MAO 1 coating. In this case, the tissue also penetrated the pores of the implant along the outer surface (Figure 7B,E). On the inner surface of the implants (from the side of the nasal cavity), there is a marginal spreading of nasal mucosa tissue onto the cylindrical part of the implants. The pores from the inner surface are not filled with soft tissues (Figure 7E). As with MAO 1 coated implants, close contact with bone tissue is observed (Figure 7D). The most pronounced connective tissue was observed on the MAO 3 coated implants (Figure 7C). This tissue penetrated deep enough into the pores of the implants along the outer surface (Figure 7F). As with MAO 2 coated implants, there is a marginal spread of the nasal mucosa tissue into the cylindrical part of the implant. Thus, the germination of soft tissue in the surface of the implants is noted. In all cases, close contact of the implant with the bone tissue is observed.

Histological evaluation: Analysis of bone tissue fragments with adjacent soft tissues from the area of implantation shows fibrosis of varying degrees observed for all 3D implants. Inflammatory changes were noted only in isolated cases, due to the absence of inflammatory inducing parameters of MAO 1–3 coatings on the 3D implants. For implants with MAO 1 and MAO 3 coatings, the level of fibrosis is moderate (Figure 7G,I), but in two cases there are small sites of productive inflammatory infiltration (MAO 1). In the the case of implants with MAO 2 coating, fibrosis is more pronounced (Figure 7H). Morphological assessment of the implantation area and the characteristics of the course of the postoperative follow-up after 60 days, the best integration properties, as well as the absence of inflammatory complications in the implantation area, were shown by 3D printed implants surface-modified with MAO 3 coatings (Figure 7G–I).

## 4. Conclusions

In this study, three different micro-arc oxidized coatings were fabricated and evaluated regarding their surface morphology, elemental composition, crystalline phase composition, electrochemical properties, cell settlement, osteogenic differentiation, as well as *in vivo* properties.

Micro-arc oxidization allows for the creation of both dense oxide-based coatings (MAO 1) and amorphous calcium phosphate (MAO 2 and MAO 3) coatings, depending on the composition of the electrolytes used. A high calcium to phosphate ratio in the coating (MAO 3) allows the fabrication of layered structured coatings with a similar structure to MAO 1 coatings, but with significantly more pores. Of all fabricated coatings, MAO 3 exhibits the highest calcium and phosphate content, while MAO 2 displays the second highest phosphate content. All three MAO coating types exhibit large fractions of amorphous content, but the MAO 2 coating has the highest amount of the amorphous phase.

The coating furthermore prevents corrosion current flow for all coatings. X-ray CT studies revealed a non-uniform coating growth on the surface of complex-shaped 3D printed products during the MAO formation process. This finding proves the need for novel quality control methods for additive manufactured complex structures.

In cell and animal studies, the MAO 3 coating proved to be superior, especially compared to the MAO 2 coating, which displayed in a few cases inflammation in addition to granulomas. Such inflammations could, in the near future, be prevented in combination with other novel systems [66,67,68]. Furthermore, the alkaline phosphatase activity was the highest for MAO 3, implying the highest rate of osteogenic differentiation. Compared to the MAO 1 coating, the anchoring, inflammation properties and cell density were superior for the MAO 3 coating. These results significantly support further improvement and modification of additively manufactured implants. Especially novel coating types can significantly improve biocompatibility and patient welfare. A high calcium to phosphorus ratio in implant coatings is extremely important for the long-term anchoring of the implant in natural bone tissues.

## Figures and Tables

**Figure 1 jfb-13-00285-f001:**
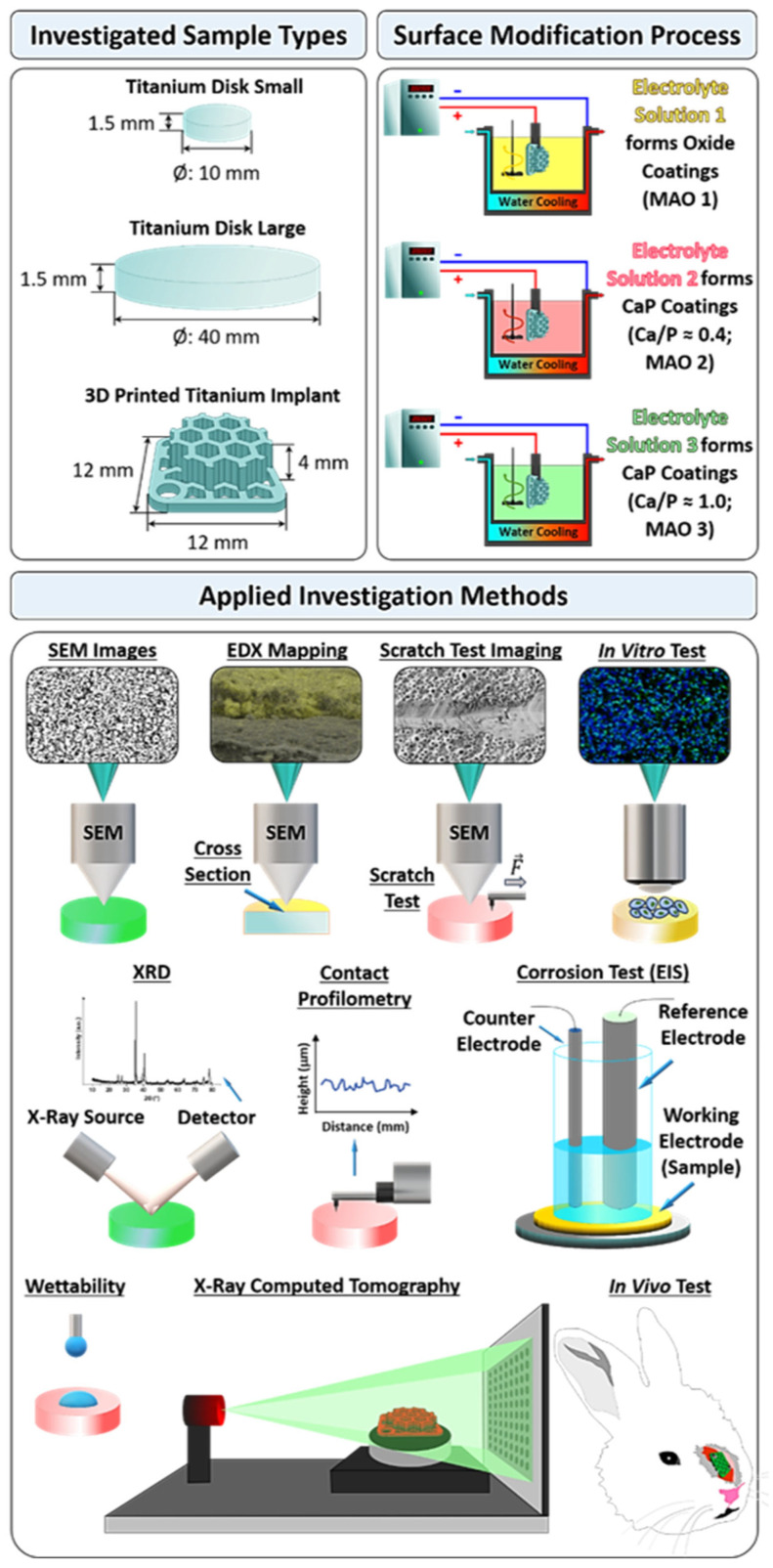
Schematic overview of the titanium sample types used in this study, schematic illustration of the surface modification via micro-arc oxidation (MAO) with three different electrolyte solutions (yellow represents the MAO 1 coating, which is a titanium oxide coating; red represents the MAO 2 coating, which is a calcium phosphate (CaP) coating; and green represents the MAO 3 coating, which is also a CaP coating), as well as schematic depictions of the utilized measurement principles and measurement setups applied in this study, such as: scanning electron microscope (SEM) imaging, energy dispersive X-ray (EDX) mapping, scratch test, contact profilometry, X-ray diffraction (XRD), corrosion test via electrochemical impedance spectroscopy (EIS), X-ray computed tomography (X-ray CT), *in vitro* tests and *in vivo* study. It should be noted that all sample types were also surface modified with all three different electrolyte solutions by micro-arc oxidation, and the different colored samples shown in this figure are only representative.

**Figure 2 jfb-13-00285-f002:**
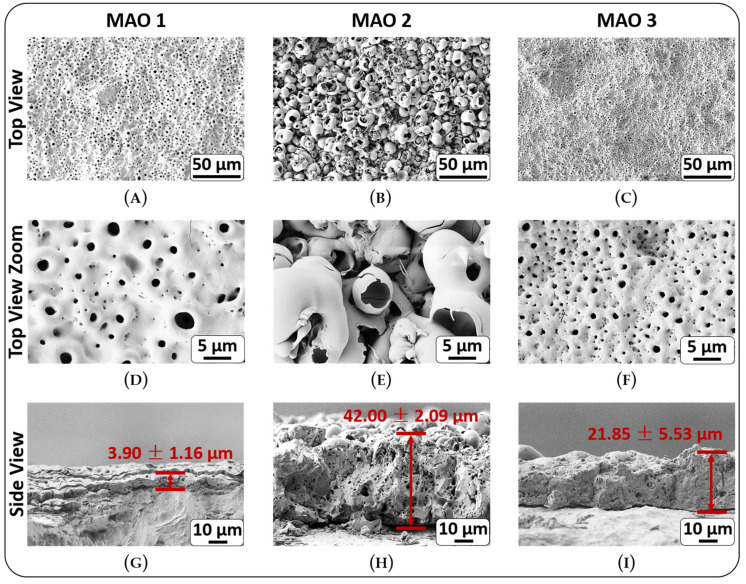
Top view SEM micrographs of the MAO 1 (**A**,**D**), MAO 2 (**B**,**E**) and MAO 3 (**C**,**F**) surface coatings formed in different electrolytes by micro-arc oxidation (MAO) and the cross-section SEM micrographs of the coating fractures with the indicated coating layer heights (**G**–**I**).

**Figure 3 jfb-13-00285-f003:**
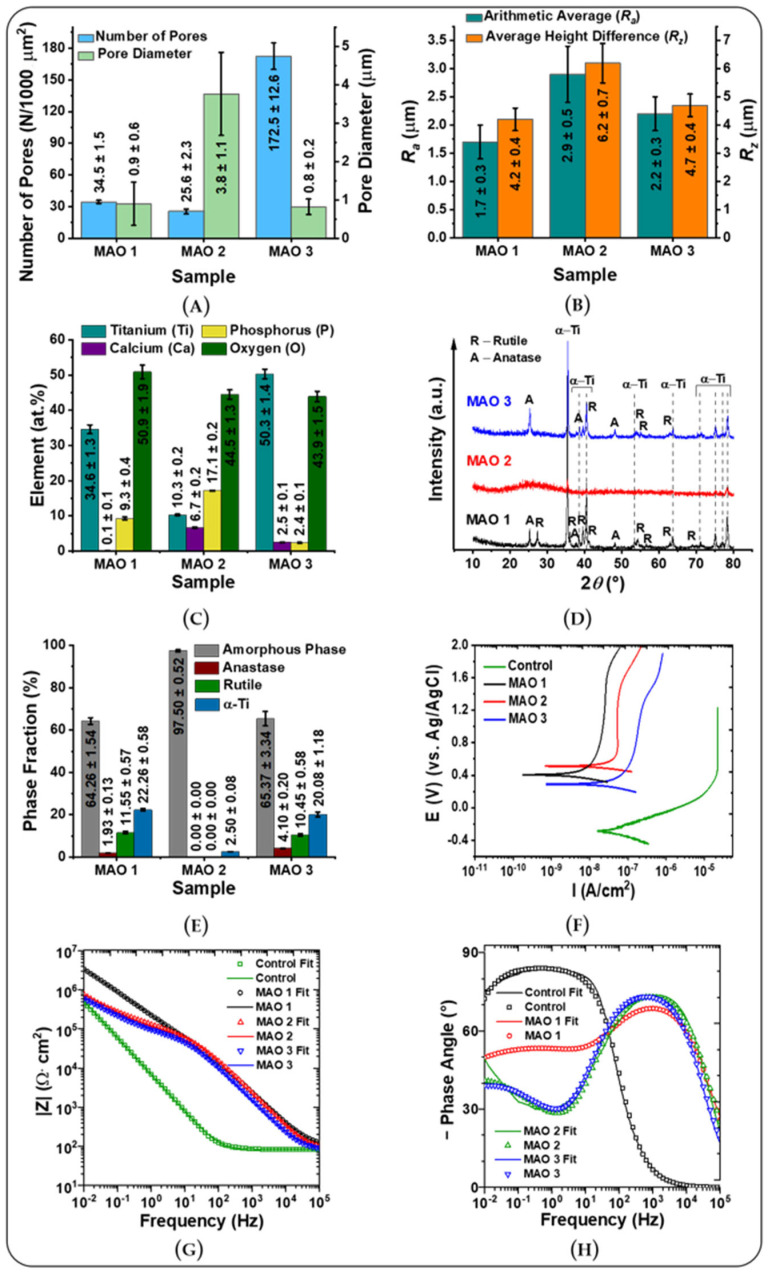
Physical and chemical properties of titanium samples coated by three different MAO coatings. (**A**) Average number of pores per surface area and average pore diameters; (**B**) surface roughness, which is shown by arithmetic average (R_a_) and average height difference (R_z_); (**C**) elemental composition determined by EDX; (**D**) X-ray diffraction pattern; (**E**) crystallite phase fraction composition calculated by peak fittings of the XRD pattern shown in (**D**); (**F**) potentiodynamic polarization curves recorded in 0.9% NaCl solution; (**G**) evolution of impedance modulus versus frequency; (**H**) evolution of phase angle versus frequency for the MAO 1–3 coatings.

**Figure 4 jfb-13-00285-f004:**
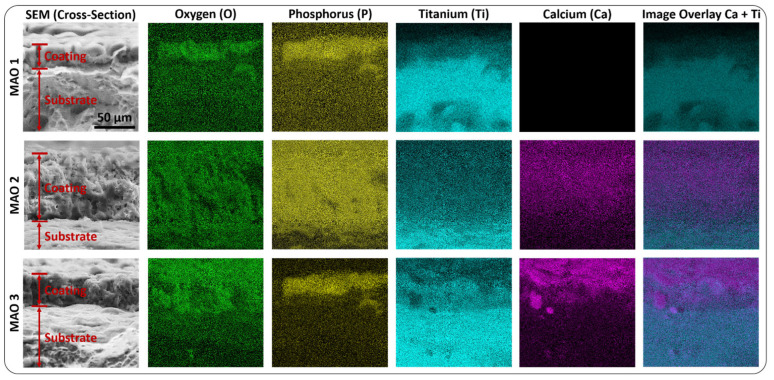
Investigation of the elemental composition of the MAO 1–3 coatings via cross-section SEM micrographs and EDX mapping. Each MAO coating reveals a significantly different layered structure. The scale bar is 50 µm and representative for all micrographs shown in this figure.

**Figure 5 jfb-13-00285-f005:**
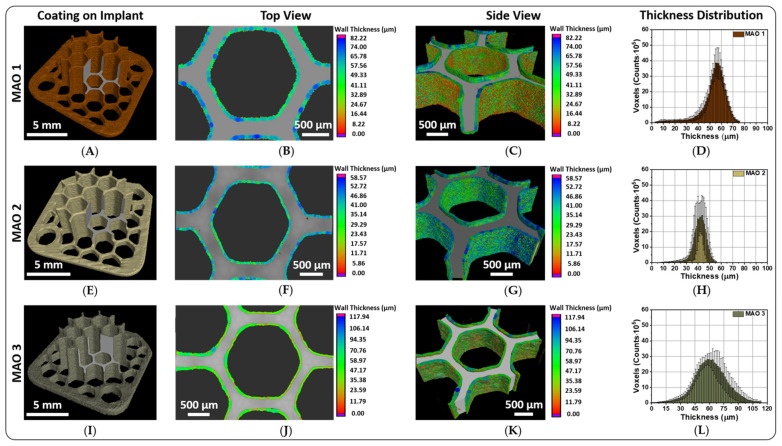
X-ray microtomography images of the 3D structured implant samples surface-modified with three different MAO coatings as full scans (**A**,**E**,**I**) and magnified scanned sections of the coatings MAO 1 (**A**–**D**), MAO 2 (**E**–**H**) and MAO 3 (**I**–**L**). The voxel size is 4.7 µm at a geometric magnification of 18× (for (**B**,**F**,**J**)) and 2.1 µm at a magnification of 40× (for (**C**,**G**,**K**)).

**Figure 6 jfb-13-00285-f006:**
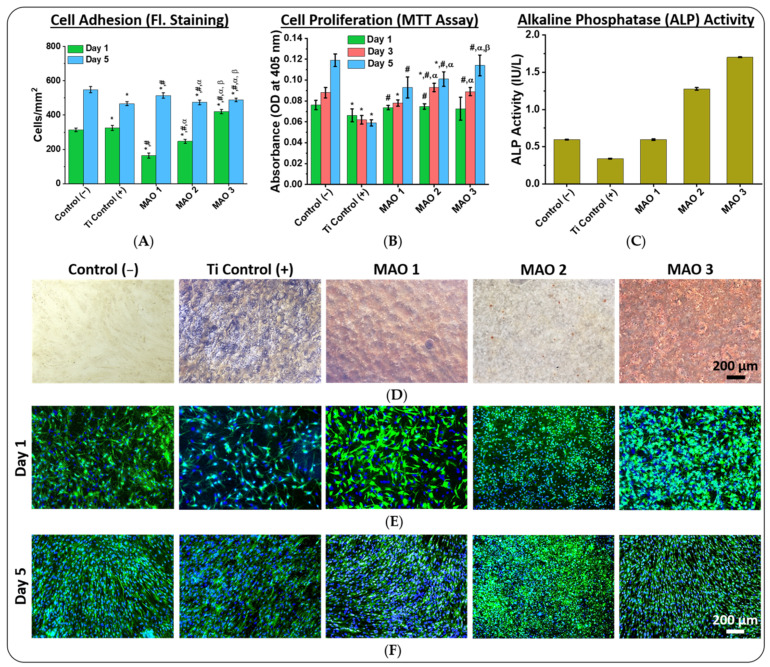
Mesenchymal stem cells’ (MSCs) growth properties on sample surfaces of plastic Petri dishes (control (−)), unmodified titanium disk samples (Ti control (+)) and the MAO 1—3 coatings applied on titanium disk samples. (**A**) Number of adherent cells determined by counting fluorescence-stained cells (*—compared to control (−); #—compared to Ti control (+); α—compared to MAO 1; β—compared to MAO 2, *p* ≤ 0.05). (**B**) cell proliferation investigated via MTT assay (*—compared to Ti control (+), α—compared to MAO 1, β—compared to MAO 2, *p* ≤ 0.05). (**C**) Osteogenic differentiation evaluated by measuring the alkaline phosphate (ALP) activity of the cells after 20 days of cell cultivation. (**D**) Bright-field micrographs of adherent cells stained with Alizarin Red after 21 days of cell cultivation. (**E**) Fluorescence micrographs of adherent cells taken on day 1 and (**F**) fluorescence micrographs of cells taken on day 5 of cell cultivation. The displayed scale bars of 200 µm are representative for all micrographs.

**Figure 7 jfb-13-00285-f007:**
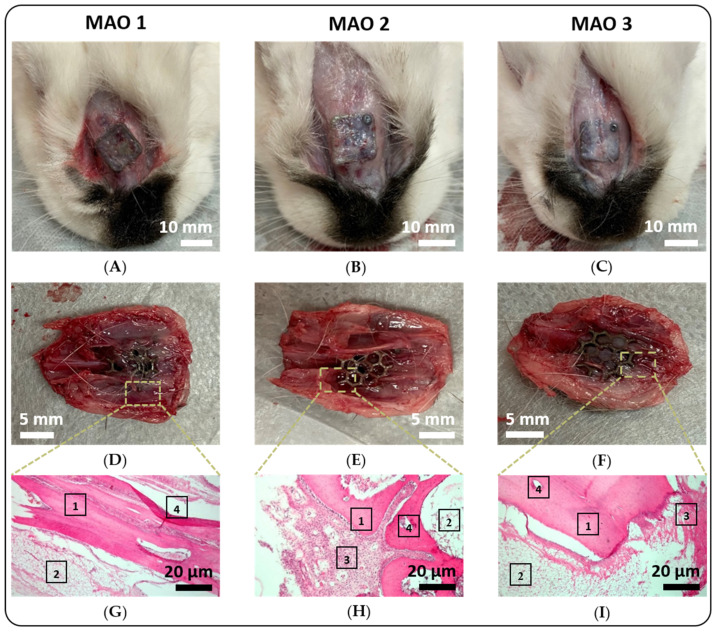
Investigation of implant anchoring in surrounding tissue. Post-implantation on day 60 and resection of implants from euthanized rabbits, appearance of the implantation site (**A**–**C**) and corresponding photographs of the extracted implant with fragments of surrounding bone tissues (**D**–**F**), as well as histological micrographs displaying a cross-section of these tissues with hematoxylin and eosin staining at a microscopic magnification of 200× (**G**–**I**). In (**G**–**I**): Position 1 refers to bone tissue below the implants, 2 shows neighboring tissue, 3 is indicating granulated tissue between the implant features, 4 shows delaminated sites due to implant removal.

## Data Availability

Data underlying the results presented in this paper are not publicly available at this time but may be obtained from the authors upon reasonable request.

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
