# Peer review of "Surface Modification of Additively Fabricated Titanium-Based Implants by Means of Bioactive Micro-Arc Oxidation Coatings for Bone Replacement"

_jfb, 2022, doi:10.3390/jfb13040285_

Round 1
Reviewer 1 Report
The study presents some interesting aspects of calcium phosphate coatings obtained by micro-arc oxidation on 3D printed Ti6Al4V alloy. Nevertheless, there are some observations that can be found below:
1. The impersonal expressions are desirable in the manuscript (e.g. “we compared”)
2. “supporting information” are not provided! The file downloaded is manuscript in editable version. I try to download from: www.mdpi.com/xxx/s1 without success.
3. If the 3D implant samples were made by a company, where the other type of samples were manufactured?
4. Some of surface characterization technique (e.g. roughness, indentation, ..) are very difficult and often impossible to apply in the case of present types of 3D printed samples.
5. There are not details about the coating process (temperature, electrolyte chemical composition, pH, …). Furthermore, coating is commonly used for a material there are deposited on other material type. In the case of oxide (first sample codification) the oxide was grown and it already existing on the surface of the alloy.
6. The PARSTAT MC is a chassis for different Potentiostat. Please provide the Potentiostat used.
7. Wy in the electrochemical experiments was not used media that simulate a human body liquid/fluid?
8. Should be provided the range of potential applied in electrochemical experiments.
9. The Tafel slopes are not recorded (are identified).
10. The order of electrochemical techniques used in experiments are wrong. The experiments must be performed in this order: OCP, LPR, Tafel, EIS.
11. The thicknesses of a coating decide it`s properties. Due to the different thickness of coatings obtained it's not good to compare them (mostly in terms of electrochemical behavior and the biocompatibility). The coating acts as a kinetic barrier in the migration of the alloy ions and then the thickness is particularly important!
12. Kipping in mind the SEM images it is difficult to assign the “dense” attribute to the coatings.
Author Response
Dear Reviewer,
we thank you for your helpful comments, which helped us to improve our article. We carefully revised the manuscript according to your comments. All changes in the manuscript has been marked in green.
Sincerely,
Johannes Frueh, Sven Rutkowski and Sergei I. Tverdokhlebov
Reviewer's comment: The study presents some interesting aspects of calcium phosphate coatings obtained by micro-arc oxidation on 3D printed Ti6Al4V alloy. Nevertheless, there are some observations that can be found below:
Reviewer's comment # 1: The impersonal expressions are desirable in the manuscript (e.g. “we compared”)
Answer: We agree with the opinion of the reviewer. Therefore, we rewrote the sentences in style of impersonal expressions in the main manuscript and the supplementary material manuscript.
Reviewer's comment # 2: “supporting information” are not provided! The file downloaded is manuscript in editable version. I try to download from: www.mdpi.com/xxx/s1 without success.
Answer: We are sorry about this mistake. We had problems with the submission system and the submission was finished before we were able to change it. The submitting author was using MDPI for the first time. Therefore, we will provide and upload the revised supplementary information manuscript after the revision to the journal system.
Reviewer's comment # 3: If the 3D implant samples were made by a company, where the other type of samples were manufactured?
Answer: We thank the reviewer for stating out this issue. The requested information has been added to the main manuscript in chapter 2 on page 3.
Reviewer's comment # 4: Some of surface characterization technique (e.g. roughness, indentation, ..) are very difficult and often impossible to apply in the case of present types of 3D printed samples.
Answer: We thank the reviewer for mentioning this problem. In the phase of planning our experiments, we were aware of this problem. Therefore, some of the analytic measurements were done on the flat samples in the form of discs, mentioned in the Figure 1 (this figure also indicates which measurement was done on which type of sample) of the main manuscript and shown as photographs in supporting information Figures S2.
Reviewer's comment # 5: There are not details about the coating process (temperature, electrolyte chemical composition, pH, …). Furthermore, coating is commonly used for a material there are deposited on other material type. In the case of oxide (first sample codification) the oxide was grown and it already existing on the surface of the alloy.
Answer: We thank the reviewer for this pointing out this issue. The coating process details can be found in the supporting information chapter “Supplementary Materials and Methods”. In the case of oxide (first sample codification): The initial oxide layer on the surface of valve metal is 5 – 10 nm. Before the formation of coatings, all samples were subjected to chemical etching in order to clean the surface of the samples and remove the oxide layer. Further, in a water-based electrolyte solution with 10% phosphoric acid (H3PO4), we formed a thick oxide coating with a thickness of 3.90 ± 1.16 μm by micro-arc oxidation.
Reviewer's comment # 6: The PARSTAT MC is a chassis for different Potentiostat. Please provide the Potentiostat used.
Answer: We thank the reviewer for this comment. PARSTAT MC is a newer model of the VersaSTAT MC that we used. We clarified the experimental part as follows: Electrochemical properties of the surface layers were investigated by an electrochemical measurement station (VersaSTAT MC, composed of 4 VersaSTATs 3 with FRA options, Princeton Applied Research, Oak Ridge, Tennessee, USA). This information has been added to chapter 2 on page 4 of the main manuscript.
Reviewer's comment # 7: Why in the electrochemical experiments was not used media that simulate a human body liquid/fluid?
Answer: We thank the reviewer for this question. A 0.9% NaCl solution is generally used as the corrosive solution for study of the degradation behavior of biocompatible materials [1]. Despite the various simulated physiological media for corrosion testing, 0.9 % NaCl was chosen as a basic testing electrolyte. This solution is consistent with the osmotic pressure of the human plasma, and only Na+ and Cl− exist in the solution. In work [2] it was suggested that 0.9 % NaCl chloride solution imitates a severe chloride environment in the body and appeared to be suitable for basic comparison tests of the titanium alloys. SBF vs MEM solutions are usually used for estimation of the influence of coatings on the bone tissue growth. A sentence about this point has been added to chapter 2 on page 4 of the main manuscript.
Reviewer's comment # 8: Should be provided the range of potential applied in electrochemical experiments.
Answer: We thank the reviewer for pointing out this lack of information. The information has been added on to chapter 2 on page 4 of the main manuscript.
Reviewer's comment # 9: The Tafel slopes are not recorded (are identified).
Answer: We thank the reviewer for stating out this issue. The values of the Tafel slopes are now added to the supporting information Table S8 on page S-12.
Reviewer's comment # 10: The order of electrochemical techniques used in experiments are wrong. The experiments must be performed in this order: OCP, LPR, Tafel, EIS.
Answer: We thank the reviewer for this statement. The experimental part has been rewritten according to the correct order of experiments performed. This information can be found in chapter 2 on page 4 of the main manuscript.
Reviewer's comment # 11: The thicknesses of a coating decide it`s properties. Due to the different thickness of coatings obtained it's not good to compare them (mostly in terms of electrochemical behavior and the biocompatibility). The coating acts as a kinetic barrier in the migration of the alloy ions and then the thickness is particularly important!
Answer: We thank the reviewer for this statement. Please let us explain our view on this point. We specifically chose these coating thicknesses because we tried to bring our experiment closer to real cases of practical application. In real clinical practice, the thicknesses of coatings based on oxides and calcium phosphates on the surface of medical implants are different! As rule, the thickness of oxide coatings on medical implants ranges from 5 to 10 µm, while the thickness of MAO-formed calcium phosphate coatings typically ranges from 20 to 40 µm. Moreover, after analyzing the literature, as well as based on the clinical application of our coatings (MAO 1, MAO 2), we have chosen the most used electrolyte compositions and modes of coating formation. In addition, the most important parameter about our coatings is not the thickness, but their chemical composition and surface morphology. The thinnest coatings show the best corrosion resistance and conversely the thickest coatings show the least protection. In comparison, the coatings show significant differences in their homogeneity. In particular, the MAO 2 coatings are full of voids and mathematically removing the voids, assuming no voids, the ”homogeneous” coating thickness would be significantly smaller (see Figure 2 main manuscript). On the one hand, the thinnest coating, the least bioactive and the most corrosion-resistant can remain in the body for a long time and would not corrode, but also not stimulate the process of new bone formation and thus better anchoring of the implant with the patient's bone tissue. On the other hand, the thickest, most soluble and bioactive coating, characterized by the lowest corrosion resistance, facilitate rapid replacement by new bone tissue during coating dissolution and tissue integration. After dissolution of such a coating, there will be no question about corrosion resistance, since the coating has been replaced by bone tissue.
Reviewer's comment # 12: Kipping in mind the SEM images it is difficult to assign the “dense” attribute to the coatings.
Answer: We thank the reviewer for this comment. Please let us explain our insights on this point. In agreement to the answer of afore comment, the analysis of SEM images (Figure 2 (G-I)) allows us to speak about the absence of pores inside the coatings of the MAO 1 group, and their presence in the coatings of the MAO 2, MAO 3 groups. In this case we observed the phenomena, that the thicker the coatings the larger the detected pores. Moreover, other researchers are actively using the SEM study to assign the “dense” or "porous" attribute to the coatings. For example, the authors of [3] use the SEM experiments to characterize porous aluminum oxide obtained by anodization of thin aluminum films. In [4] the classification of dense and porous structure of Au/Se/porous TiO2/compact TiO2/fluorine-doped thin oxide-coated glass plates of was carried out using SEM. The porous and dense structures and also the measurement of the dense layer using analyzing of the SEM images of polymeric membranes were carried out in [5].
References:
- Gnedenkov, A. S.; Sinebryukhov, S. L.; Filonina, V. S.; Egorkin, V. S.; Ustinov, A. Y.; Sergienko, V. I.; Gnedenkov, S. V. The detailed corrosion performance of bioresorbable Mg-0.8Ca alloy in physiological solutions. J. Magnes. Alloy. 2022, 10, 1326–1350, doi:10.1016/j.jma.2021.11.027.
- Rondelli, G.; Vicentini, B. Effect of copper on the localized corrosion resistance of Ni–Ti shape memory alloy. Biomaterials 2002, 23, 639–644, doi:10.1016/S0142-9612(01)00142-9.
- Gâlcă, A. C.; Kooij, E. S.; Wormeester, H.; Salm, C.; Leca, V.; Rector, J. H.; Poelsema, B. Structural and optical characterization of porous anodic aluminum oxide. J. Appl. Phys. 2003, 94, 4296–4305, doi:10.1063/1.1604951.
- Nguyen, D.-C.; Tanaka, S.; Nishino, H.; Manabe, K.; Ito, S. 3-D solar cells by electrochemical-deposited Se layer as extremely-thin absorber and hole conducting layer on nanocrystalline TiO2 electrode. Nanoscale Res. Lett. 2013, 8, 8, doi:10.1186/1556-276X-8-8.
- Alqaheem, Y.; Alomair, A. A. Microscopy and Spectroscopy Techniques for Characterization of Polymeric Membranes. Membranes (Basel). 2020, 10, 33, doi:10.3390/membranes10020033.

Reviewer 2 Report
I have reviewed the manuscript jfb-2041482 entitled "Surface modification of additively fabricated titanium-based implants by means of bioactive micro-arc oxidation coatings for bone replacement" for publication in the Journal of Functional Biomaterials. The results reported in this paper could be valuable for publication. However, several points have to be reconsidered for minor revision. In my opinion, the following points have to be considered:
(1) The authors did not explain the novelty and significance of their work in the introduction section; there are several studies about the MAO treatment of AM’ed Ti scaffolds. Moreover, this section is not cohesive. Indeed, this section is intended to "convey the core findings of the paper", i.e. reflect the best novelty of this paper in a concise form. The authors shall show the best novelty of the work, such as how your research advances the state-of-the-art of the topic/area, and /or how much better is your work compared with peer researchers on the same or similar topics. At the end of this section, the main objective of this study must be mentioned.
(2) Source of Ti64 alloy should be provided. Also, a source of chemical materials should be provided. If the homemade power supply was used for MAO how one could ensure the reproducibility of the results? Many essential details of surface characterization and evaluation are missing. The description is virtually reduced to listing the methods and instruments used, rather than details of procedures and subsequent analysis.
(3) The negative sign (-) has been missing from the y-axis of the EIS Bode phase plot in Fig. 3F.
(4) On page 12, lines 442-443, the sentence “Corrosion test: Since the biomedical application … and phase composition.” needs the following reverence: Materials Chemistry and Physics 276 (2022): 125436.
(5) There are some errors in the Figure captions. Please revise them.
(6) In the manuscript, the authors sometimes use x/y, and sometimes x.y-1 and x y-1 and xy-1, please check and choose the unified unit for the values. For example, A/cm2, mm/year, and F cm−1.
Author Response
Dear Reviewer
we thank you for your helpful comments, which helped us to improve our article. We carefully revised the manuscript according to your comments. All changes in the manuscript has been marked in green.
Sincerely,
Johannes Frueh, Sven Rutkowski and Sergei I. Tverdokhlebov
Reviewer's comment: I have reviewed the manuscript jfb-2041482 entitled "Surface modification of additively fabricated titanium-based implants by means of bioactive micro-arc oxidation coatings for bone replacement" for publication in the Journal of Functional Biomaterials. The results reported in this paper could be valuable for publication. However, several points have to be reconsidered for minor revision. In my opinion, the following points have to be considered:
Reviewer's comment # 1: The authors did not explain the novelty and significance of their work in the introduction section; there are several studies about the MAO treatment of AM’ed Ti scaffolds. Moreover, this section is not cohesive. Indeed, this section is intended to "convey the core findings of the paper", i.e. reflect the best novelty of this paper in a concise form. The authors shall show the best novelty of the work, such as how your research advances the state-of-the-art of the topic/area, and /or how much better is your work compared with peer researchers on the same or similar topics. At the end of this section, the main objective of this study must be mentioned.
Answer: We thank the reviewer for stating out this issue. We revised the introduction.
Reviewer's comment # 2: Source of Ti64 alloy should be provided. Also, a source of chemical materials should be provided. If the homemade power supply was used for MAO how one could ensure the reproducibility of the results? Many essential details of surface characterization and evaluation are missing. The description is virtually reduced to listing the methods and instruments used, rather than details of procedures and subsequent analysis.
Answer: We thank the reviewer for this question and comment. We used titanium alloy Ti6Al4V conforming to ASTM F136 (Standard Requirements for Wrought Titanium-6-Aluminum-4-Vanadium Ultrafine Grain Alloy for Surgical Implant Applications (UNS R56401)) as substrates for coating formation. The sources of the titanium alloy and chemical materials are provided in the support information chapter “Supplementary Materials and Methods”. The description in the main manuscript is reduced in order to avoid an overloading of the main manuscript. Therefore, we used a support information manuscript. The used MAO power supply is a professional device, which has been built from the company MICROSPLAV LTD., Tomsk, Russia and for which we have a patent (details of the pulsed power supply is given in reference 39). MAO setup for the formation of MAO coatings was developed by TPU together with the MICROSPLAV LTD Company within the framework of the Federal target program for the production tasks of the industrial partner «Osteomed-M» LLC (Moscow, Russian Federation). MICROSPLAV LTD is an organization producing specialized high-current power supplies that are imported abroad (https://www.microsplav.ru/about/). To control the reproducibility of the results, we study the composition and structure of the formed coatings every time in the same modes. Similar power sources have already been used by us to form bioactive coatings on the surface of medical implants for hospitals all over Russia since a few years. Therefore, we can ensure the reproducibility of the results. Otherwise, we would not be requested by our medical partners repeatedly.
Reviewer's comment # 3: The negative sign (-) has been missing from the y-axis of the EIS Bode phase plot in Fig. 3F.
Answer: We thank the reviewer for finding this issue. A negative sign (-) has been added to the y-axis of the EIS Bode phase plot in Figure 3H.
Reviewer's comment # 4: On page 12, lines 442-443, the sentence “Corrosion test: Since the biomedical application … and phase composition.” needs the following reverence: Materials Chemistry and Physics 276 (2022): 125436.
Answer: We thank the reviewer for the suggestion of this useful reference. The reference has been added as reference 28.
Reviewer's comment # 5: There are some errors in the Figure captions. Please revise them.
Answer: We thank the reviewer for this comment. All figure caption has been checked again.
Reviewer's comment # 6: In the manuscript, the authors sometimes use x/y, and sometimes x.y-1 and x y-1 and xy-1, please check and choose the unified unit for the values. For example, A/cm2, mm/year, and F cm−1.
Answer: We carefully checked the whole manuscript. We use only the x/y in the whole manuscript.

Reviewer 3 Report
Dear authors,
please find attached my comments.

Author Response
Dear Reviewer,
we thank you for your helpful comments, which helped us to improve our article. We carefully revised the manuscript according to your comments. All changes in the manuscript has been marked in green.
Sincerely,
Johannes Frueh, Sven Rutkowski and Sergei I. Tverdokhlebov
Reviewer's comment: The manuscript by Kozelskaya et al. reports surface modification of additively fabricated titanium-based implants using bioactive micro-arc oxidation coatings for bone replacement. The manuscript is well structured and covers a very interesting topic. After some suggestions, in my opinion, the following manuscript can be published in JFB.
Reviewer's comment # 1: Please unify the bibliographical references. Some references are separated by commas, others by hyphens. Please choose which one to use.
Answer: We thank the reviewer for this comment. It is a standard procedure to precent references in the style we made in paper. For example: … modification of the composition of the electrolyte solution and increasing the power of electrical equipment [22,30–32]. The references 30, 31 and 32 are part of this citation and thus merged to 30–32. As reference 22 is also suitable to be cited here, reference 22 is separated by a comma from the merged references. Therefore, a merging to 22 – 32 would be incorrect, because the references 23, 24, 25, 26, 27, 28 and 29 are not part of this citation as they are not suitable here.
Reviewer's comment # 2: The introduction should be strengthened. The authors could compare these titanium-based implants with others in the literature, (e.g. 10.1016/j.matdes.2021.110350;10.3390/ma15093283;10.1002/smll.202201869 and so on).
Answer: We thank the reviewer for this helpful comment. The introduction has been revised.
Reviewer's comment # 3: According to the ASTM, what type of commercially pure titanium was used?
Answer: We thank the reviewer for this comment. We used titanium alloy Ti6Al4V conforming to ASTM F136 (Standard Requirements for Wrought Titanium-6-Aluminum-4-Vanadium Ultrafine Grain Alloy for Surgical Implant Applications (UNS R56401)) as substrates for coating formation. This information has been added to the Materials and Methods section on page 3 of the main manuscript.
Reviewer's comment # 4: How the surface of titanium discs was prepared before the functionalization? Did the authors perform any polishing procedures? How was the surface topography standardized?
Answer: We thank the reviewer for this comment. The surface preparation procedure was carried out in the way that all initial samples were characterized by the same surface roughness after chemical etching and cleaning. The description of the preparation procedure has been added to the Supplementary Materials and Methods section on page S-2 of the supplementary information manuscript.
Reviewer's comment # 5: Did the authors carry out wettability tests? As is well known, among the various surface properties, hydrophilicity is closely related to cell adhesion, as cell proliferation and differentiation have been shown to increase on the surface of highly hydrophilic materials.
Answer: We thank the reviewer for this comment. According to reviewer comment, we carried out wettability tests and calculated the surface energies. The results were added to supplementary materials manuscript as SI Figure S4 (page S-5) and SI Table S2 (page S-9). In addition, the method description has been added to the main manuscript on pages 3 and 4. In chapter 3, the results has been discussed on pages 9, 15 and 16.

Round 2
Reviewer 1 Report
Even that some improvements can be made, In this form the manuscript can be published.